

# Disintegration half-life of biodegradable plastic films on different marine beach sediments

Andreas Eich, Miriam Weber and Christian Lott

HYDRA Marine Sciences GmbH, Bühl, Germany

Corresponding author
Christian Lott,
c.lott@hydramarinesciences.com

## ABSTRACT

The seafloor is considered the major sink for plastic debris in the world's oceans. Biodegradable polymers are available on the market as a substitute for conventional plastic and could potentially end up in the same environment. To gain more insight into the effects of different sediments on the degradation rate of biodegradable plastic we performed two iterative seawater tank experiments. First, to test the effect of sediment grain size, film of Mater-Bi HF03V, a blend of thermoplastic starch and biodegradable polyesters, was placed on the surface of mud as well as on four different grain size fractions of beach sand. Disintegration half-life was shortest on mud (139 days) and increased with the grain size of the beach sediment fractions (63–250 μm: 296 days; 250–500 μm: 310 days; 500–1,000 μm: 438 days; >1,000 μm: 428 days). We assume that the higher surface-to-volume ratio in fine sediment compared to coarse sediment led to a higher bacterial abundance and thus to faster disintegration rates. In a follow-up experiment, the <500 μm fraction of sediment from four different beaches around Isola d'Elba, Italy, was used to test plastic disintegration as above. Additionally, polyhydroxybutyrate (PHB, MIREL P5001) was used as a positive control and high-density polyethylene (HD-PE) as a negative control. No disintegration was observed for HD-PE. Mater-Bi HF03V and PHB disintegrated significantly differently on sediment from different sites, with half-lives of Mater-Bi HF03V ranging from 72 to 368 days and of PHB from 112 to 215 days. Here, the half-life was shortest on slightly coarser sediment and at potentially anthropogenically impacted sites. We assume that the effect of the grain size on the disintegration rate was masked by other parameters influencing the microbial community and activity. Understanding the parameters driving biodegradation is key to reliably report the range of disintegration rates occurring under the various conditions in different ecosystems.

## INTRODUCTION

Plastic pollution occurs in all ecosystems on the planet, including deep-sea trenches (*Fischer et al., 2015*), remote mountains (*Allen et al., 2019*), air (*Dris et al., 2017*), and the polar regions (*Greenpeace, 2018*; *Peeken et al., 2018*). Harmful effects of plastic waste in the
environment are widely known (*Li, Tse & Fok, 2016*) and range from direct adverse impact on animals and plants (*Derraik, 2002*; *Gregory, 2009*; *Guzzetti et al., 2018*) to economic costs for society (*Lee, 2015*; *McIlgorm, Campbell & Rule, 2011*).

The substitution of conventional materials with biodegradable plastic, *i.e.* polymers and blends that degrade predominantly by the enzymatic action of microorganisms into $CO_2$ (and in the absence of oxygen also $CH_4$), water, inorganic compounds, and biomass (*Albertsson et al., 2020*), are discussed for certain applications. This can be a reasonable strategy in cases where plastics are made to remain in the environment (*e.g.* geotextiles), the potential of being lost is high (*e.g.* cigarette butts, agricultural mulch film, fish boxes), or the input into the environment is unavoidable (*e.g.* abrasion of tires, shoes, or textiles). As part of an assessment of potential environmental benefits and risks, the biodegradability of a material has to be proven before raising any claim thereof (*Albertsson et al., 2020*; *Weber et al., 2018*). Whether or not biodegradation takes place at all, and if so at which rate, not only depends on the chemical structure of the material but also on the conditions in the considered environment (*Albertsson et al., 2020*), which include temperature, pH, humidity, oxygen availability, and the activity of microorganisms and their enzymes (*Laycock et al., 2017*). There is a variety of standard test methods available to assess biodegradability and biodegradation rates in different scenarios, simulating managed environments such as industrial and home compost or compartments of the open environment such as soil, freshwater, or the sea (*Harrison et al., 2018*). But even within one compartment, like the marine realm, conditions vary between different habitats, such as beaches, the water surface, the water column, the shallow-water seafloor, and the deep sea, all of which also differ by climate zone (*Lott et al., 2021*).

In laboratory tests, the biodegradability of a material can be proven by exposing the sample to marine microbial inocula under controlled conditions. Plastic samples are incubated in closed vessels, typically with a volume between 0.1 and 1 L, and the metabolic end-products are analysed. This method has the advantage that plastic biodegradation can directly be quantified by measuring the evolved $CO_2$ or the consumed $O_2$ (*Tosin et al., 2012*; *ISO 18830, 2016*; *ISO 19679, 2020*). On the other hand, test conditions closer to real nature and applicable also to real products (*e.g.* bags, packaging) are achieved by exposing the plastic samples *in-situ* or in mesocosms (tanks) with a much higher volume than the laboratory flasks (*Lott et al., 2020*). In these open-system tests, the measurement of evolving $CO_2$ is not possible, therefore plastic biodegradation cannot be directly quantified. Here, the disintegration of the sample can be measured as a proxy for the biodegradation of materials that have been proven to be biodegradable in closed-system laboratory tests, if the samples are well protected from physical impact.

Most biodegradable polymers have a specific density higher than 1 and will sink in seawater (*e.g.* 1.28 for Mater-Bi HF03V (*Tosin, Pischedda & Degli-Innocenti, 2019*), 1.30 for polyhydroxybutyrate MIREL P5001 and 1.20 for Kaneka PHBH, retrieved from www.matweb.com). Also materials with a lower specific density or items containing lighter compartments such as bubbles or foam will eventually sink due to fouling (*Pauli et al., 2017*). The marine seafloor is considered the major sink for plastic debris in the world's ocean, with an estimated share of 94% of all marine plastic accumulating (*Eunomia, 2016*).

**Table 1  Materials used in both experiments.**

| Material name | Experiment | Thickness (µm) | Sample size (cm) |
|---|---|---|---|
| Mater-Bi HF03V | Pilot | 25 | 20 × 2 |
| Mater-Bi HF03V | Follow-up | 12 | 6 × 8 |
| Polyhydroxybutyrate MIREL P5001 | Follow-up | 85 | 6 × 8 |
| HD-PE | Follow-up | 10 | 6 × 8 |

Plastic items sunken to the seafloor are initially in contact with two matrices, seawater and sediment, until they are buried by sedimentation or biodegraded.

Most of the seafloor is covered with sediment. Depending on the hydrodynamic regime at a given location the grain size ranges from blocks, stones, and gravel to sand and mud. Sediment properties as *e.g.* permeability, oxygenation, and nutrient content directly or indirectly depend on the grain size, as do the microbial community and the biogeochemical processes linked to it (*e.g. Ahmerkamp et al., 2020*). This leads to the hypothesis that the biodegradation potential of sediment is influenced by its grain size. Albeit the importance as major plastic sink, studies on the degradation of plastic at the seafloor are scare and we are not aware of any study comparing the impact of different benthic conditions, *e.g.* sediment grain size, on degradation rates.

To gain more insight into the effects of different sediments on the degradation rate of biodegradable plastic we performed two iterative tank experiments. In a pilot study, marine sediment from one location was divided into four grain size fractions by sieving. The sediment fractions and mud were incubated with seawater in aquaria and the disintegration of the biodegradable plastic film Mater-Bi HF03V (Novamont, Italy) laying on the sediment surface was measured. In a follow-up experiment, sediment was collected from four different locations and directly put into aquaria. Here, plastic disintegration was tested the same way as above. Additionally, polyhydroxybutyrate (PHB) was used as a positive control and high-density polyethylene (HD-PE) as a negative control.

# MATERIALS & METHODS

## Test materials

In both experiments, the degradation of Mater-Bi HF03V, a blend of thermoplastic starch and biodegradable polyesters, was investigated (Table 1). In the pilot experiment, this material was still in an experimental phase and was tested as a film of 25 µm thickness. In the follow-up study, the material was commercially available as fruit and vegetable bags of 12 µm thickness. The material (labelled 'F&V Ecobag') was retrieved from a local supermarket and was produced by Erreti S.r.l. (Solbiate Olona, Italy) with Mater-Bi (Novamont, Italy). Mater-Bi HF03V is certified as home compostable (OK compost HOME, Vinçotte S373, now TÜV Austria) and is biodegradable in soil (*Pischedda, Tosin & Degli-Innocenti, 2019*; *Tosin, Pischedda & Degli-Innocenti, 2019*). More information on the specific material properties can be found in *Pischedda, Tosin & Degli-Innocenti (2019)* and *Tosin, Pischedda & Degli-Innocenti (2019)*.
In the second experiment, two additional materials were tested as controls (Table 1). Polyhydroxybutyrate (PHB, MIREL P5001, Metabolix, Cambridge, USA), described by the producer as 'PHA copolymer', is a polyester that is naturally produced by different microorganisms (*Lu, Tappel & Nomura, 2009*) and is biodegradable under marine conditions (*Lott et al., 2021*). An FTIR analysis (Fig. SI-1) confirmed the material as PHB. Films of 85 μm thickness were used as a positive control to show that material disintegration is possible when using the selected matrices. High-density polyethylene (HD-PE) fruit and vegetable bags with a thickness of 10 μm were obtained from a local market (Marina di Campo, Italy). PE is the most widely used polymer type for packaging and a conventional plastic that is regarded as not biodegradable. It was used as a negative control to prove that no material disintegration occurred which is caused by physical impact.

## Experiments

Two experiments were performed in aquaria: to study the effect of sediment grain size on material disintegration, the disintegration of the biodegradable plastic film laying on different grain size fractions of beach sand, and additionally on mud, was tested in a pilot experiment. In a follow-up experiment, three plastic materials were tested on natural sediments from four different beaches.

## Pilot experiment

Natural beach sand was collected at 0.1 m water depth at the beach of Fetovaia (Isola d'Elba, Italy, Mediterranean Sea, 42° 44′ 00.1″ N, 010° 09′ 15.3″ E, Fig. 1), brought to the laboratory, and wet sieved with natural seawater through standard sieves (Retsch, Germany) to the fractions of >1,000 μm, 1,000–500 μm, 500–250 μm, and 250–63 μm. Additionally, mud was taken from the former saline basin at Terme S. Giovanni (Portoferraio, Isola d'Elba, 42° 48′ 12.1″ N, 010° 19′ 01.0″ E) and used without further treatment. Five glass aquaria (30 × 20 × 20 cm) were filled with a 6 cm thick layer of sediment (3.6 L) and a 10 cm thick layer of seawater (6 L) collected at Seccheto (42° 44′ 06.5″ N, 010° 10′ 33.5″ E, Fig. 1). The tanks were closed with plastic film to reduce evaporation. The water in each test tank was aerated and slightly stirred by a bubbler connected to an air pump at ca. 37.5 L h$^{-1}$ (SCHEGO, Germany). Aquaria were kept in a climate chamber for ten months at 21 °C (Fig. SI-2). Salinity was regularly checked and kept at 39 with distilled water to compensate for evaporation loss. Salinity, oxygen concentration, temperature, and pH were measured once every month with a conductivity sensor TetraCon 925, an oxygen sensor FDO 925, and a pH sensor SenTix 940, attached to a Multi 3420 (WTW, Weilheim, Germany).

Mater-Bi HF03V film was cut to the size of 20 × 2 cm. Five test strips were put between two layers of plastic mesh (mesh width: 4 × 4 mm, diamond-shaped, polypropylene) to prevent the potential loss of fragments and placed on the sediment surface of each test tank (Fig. 2, Fig. SI-2). After 5 months (153 days), the samples were retrieved, carefully removed from the mesh, and photographed (Canon 5D Mark II or Canon 600D with 50 mm macro lens) immersed in seawater. Afterwards, the sample strips were re-arranged
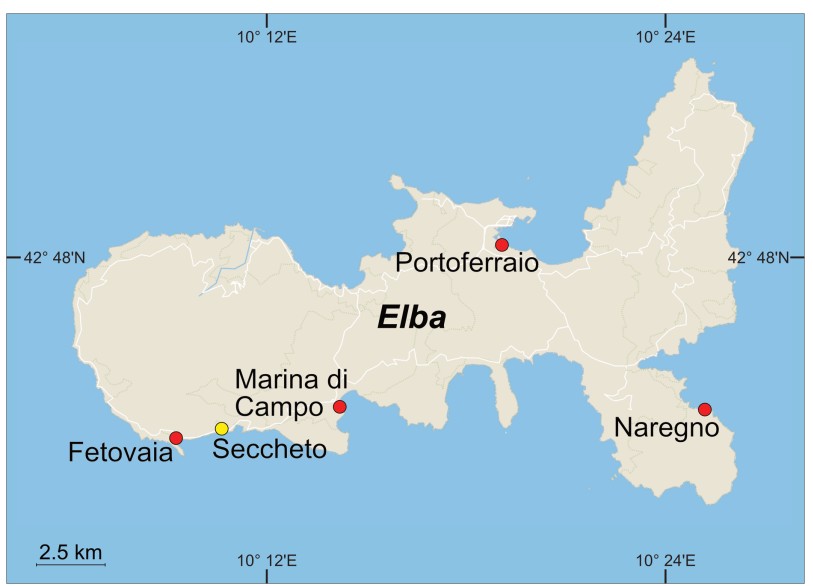

**Figure 1 Sampling sites around the Island of Elba, Italy.** *Pilot experiment*: Beach sediment was collected in Fetovaia and mud in Portoferraio. *Follow-up experiment*: Sediment was collected in Fetovaia, Marina di Campo, Portoferraio and Naregno. Seawater for both experiments was collected in Seccheto (yellow dot). ©OpenStreetMap contributors.

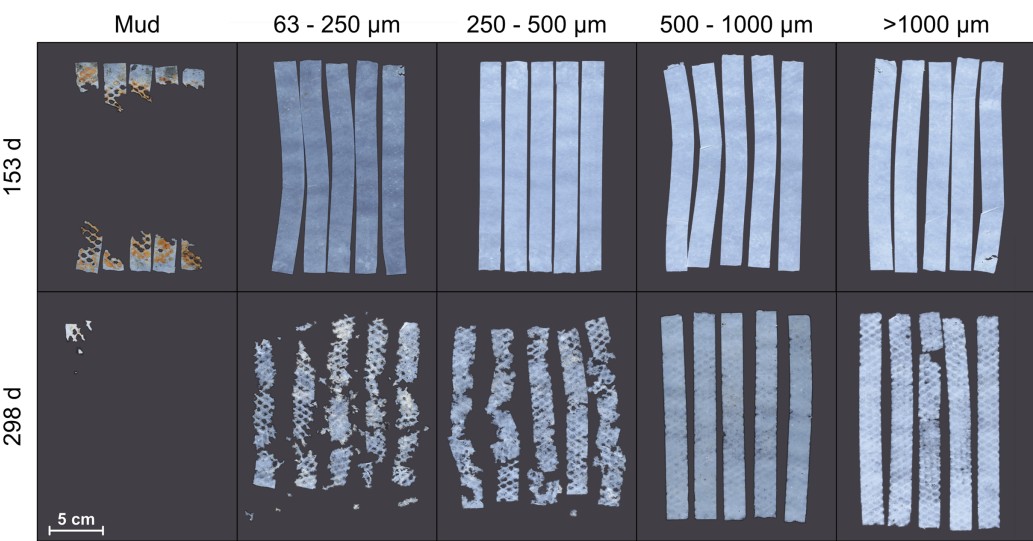

**Figure 2 Samples of Mater-Bi HF03V.** Sample strips were placed at the sediment surface of the grain size fractions >1,000 μm, 500–1,000 μm, 250–500 μm, 63–250 μm, and mud (columns) and photographed after 153 and 298 days (rows).

between the mesh and re-exposed in the tanks. After 10 months (298 days), the sampling procedure was repeated and the experiment terminated.

## Follow-up experiment

At four beaches around Isola d'Elba (Fig. 1), sediment was collected at 0.1 m water depth. In Fetovaia (same site as used for the pilot experiment), Marina di Campo (42° 44′ 36.13″ N,

10° 14′ 7.27″ E), Portoferraio (42° 48′ 21.00″ N, 10° 19′ 2.40″ E), and Naregno (42° 45′ 13.20″ N, 10° 24′1 9.80″ E), sediment was sampled as described above and wet-sieved through a 500 μm mesh to remove coarser particles and debris like shells, pebbles, and seagrass leaves.

Three glass aquaria (30 × 20 × 20 cm) were filled with a layer of 4 cm (2.4 L) of each of the four sediment types and 6 L of natural seawater collected in Seccheto (Fig. 1), which led to a replication of $n = 3$ for each sediment treatment. All 12 tanks were sealed with a lid made of plastic film with a small hole to allow the passage of an air hose (Fig. SI-4).

The salinity was regularly checked as described above and if necessary adjusted to 39 with distilled water. Dissolved oxygen was around 95% air saturation and pH around 8.1 (Fig. SI-5). In contrast to the pilot experiment, the aquaria were not kept in a climate chamber but the temperature was allowed to change with the seasons. The water temperature was monitored with loggers (UA-002-64, Onset Computer Corporation, Bourne, USA) and ranged from 12 to 29 °C (Fig. SI-6). Samples from the four sediment types and natural seawater used for the experiments were sent to an external laboratory for grain size analysis (CSA, Rimini, Italy) and basic geochemical characterization (Limnowak, Ottersberg, Germany). The grain size distributions were analysed using GRADISTAT V8 (*Blott & Pye, 2001*). The organic matter, calcium-carbonate, and silicate contents were calculated using the loss-on-ignition method (*Heiri, Lotter & Lemcke, 2001*).

Test materials were cut to strips of 6 × 8 cm. Each specimen was put between two layers of polyester mesh (SEFAR, Heiden, Switzerland) with a square-shaped mesh of 2 × 2 mm and held together at the short sides with plastic clamps. One replicate of each material was randomly placed in each aquarium.

After the start of the experiment, sampling occurred after 129, 186 and 252 days for the experiments with Marina di Campo, Portoferraio, and Naregno sediment. The last sampling for the test with Fetovaia sediment was after 356 days because after 252 days, the disintegrated area of Mater-Bi HF03V in the Fetovaia treatment was estimated to be below 50% and much lower than in the other treatments. During sampling, one replicate of each material was retrieved from each of the three aquaria containing the same type of sediment. The upper mesh was carefully removed. Depending on the state of disintegration, the samples were photographed with or without the lower mesh. No re-exposure of the specimens occurred.

## Determination of disintegration

Disintegration was determined photogrammetrically as described before (*Lott et al., 2020*). The remaining plastic was marked on the photos and the area loss of the total exposed area was measured using ImageJ (*Abramoff, Magalhaes & Ram, 2004*).

## Statistical analysis–calculation of half-life

All data was analysed in R (*R Core Team, 2020*) as described in *Lott et al. (2021)*. The disintegration over time was modelled using beta regression (*betareg* package) and the appropriate link-function (logit, cloglog, cauchit, or loglog) selected by comparing the Root Mean Square Deviation (package *caret*). Half-life was calculated by

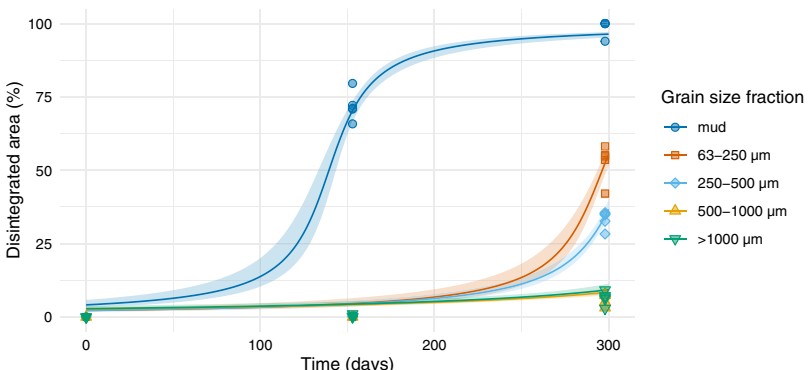

**Figure 3** **Disintegration of Mater-Bi HF03V.** Film (25 µm) was exposed on different grain size fractions of the same sediment and mud. Model curve and 95% confidence interval (shaded area).

back-transforming link-functions for 50% material disintegration and solving the formula for the time ($t_{0.5}$) using the coefficients estimated by the linear model. Monte-Carlo simulations were used to calculate 500,0000 half-live values considering the variance of the model coefficients, which allowed statistical tests with empirical $p$-values.

# RESULTS

## Pilot experiment

The test material Mater-Bi HF03V showed substantial disintegration on mud after 153 days and on all grain size fractions within 298 days (Fig. 2).

The half-life differed significantly between different sediment size fractions and was lowest for samples on mud (139 days), followed by samples on the sediment with a size fraction of 63–250 µm (296 days), 250–500 µm (310 days), and sediment above 500 µm (Figs. 3 and 4, Tables SI-1 and SI-2). Throughout the experiment, the monitored environmental conditions (dissolved oxygen concentration, temperature, pH, and salinity) varied little (Fig. SI-3).

## Follow-up experiment

Disintegration was obvious for the biodegradable materials Mater-Bi HF03V and PHB (Fig. 5, Fig. SI-7). HD-PE did not disintegrate within 356 days.

The predicted half-life for Mater-Bi HF03V ranged from 72 to 368 days and for PHB from 112 to 259 days (Fig. 6, Table SI-4). PHB disintegrated significantly faster on Marina di Campo sediment than on all other sediment types. Also for Mater-Bi HF03V, the disintegration on Marina di Campo sediment and on Portoferraio sediment was significantly faster than on Naregno sediment (Tables SI-5 and SI-6). The observed disintegration of Mater-Bi HF03V on Fetovaia sediment and of PHB on Naregno sediment was variable at the last sampling point (Mater-Bi HF03V after 356 days: 0%, 54.8%, and 97.8% area loss; PHB after 252 days: 7.1%, 50.3%, and 88.3% area loss).

All sediment types consisted almost completely of silicate (97–98%, Table 2). According to the *Folk & Ward (1957)* method, the sediment collected in Marina di Campo was

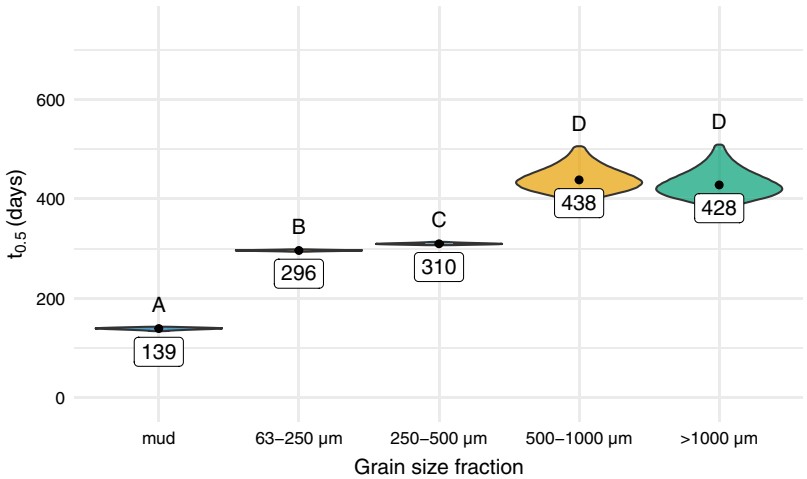

**Figure 4 Predicted half-life of Mater-Bi HF03V film (25 μm) on different grain size fractions of the same sediment and mud.** The distributions of the 95% confidence intervals of 500,000 calculated half-life values are shown as violin plots. Significantly different half-life values (empirical *p*-values) have different letters above the plots. The number below the point is the predicted half-life.

medium sand, all other sediments were fine sand. All sediment types were well or very well sorted (Fig. 7, Table 2). Portoferraio sediment had a higher phosphorous content (190 mg kg$^{-1}$ dry weight) than the other sediments (80–110 mg kg$^{-1}$ dry weight, Table 2). No statistical tests could be performed as the analysis was done for only one sample per site. In the water phase of the aquaria, environmental conditions such as salinity, pH, and oxygen saturation varied little throughout the experiment (Fig. SI-5). The concentration of nitrate was 0.14 mg L$^{-1}$, of ortho-phosphate 0.008 mg L$^{-1}$, and of ammonia 0.13 mg L$^{-1}$ (Table SI-3).

## DISCUSSION

In the pilot experiment, it was shown that the disintegration rate of biodegradable plastic film on marine sediment is affected by grain size. Disintegration time was shortest on mud, followed by samples laying on the smallest grain size fraction and significantly increased with increasing grain size of the sieved sand (Fig. 4, Tables SI-1 and SI-2). The biodegradation rate of organic substances depends on the activity and abundance of microorganisms. Both are influenced by environmental conditions such as temperature, pH, salinity, and oxygen concentration. Sediment characteristics, like grain size, porosity, and permeability influence these conditions (*Ahmerkamp et al., 2020*; *Probandt et al., 2017*). Permeability increases with grain size. High permeability is usually connected with low bacterial biomass (*Hou et al., 2017*) and thus, hypothetically, with a lower biodegradation rate. However, if plastic samples are laying on the sediment surface, the permeability of the underlaying matrix may only play a minor role.

   Most benthic bacteria live attached to sediment grains (*Probandt et al., 2018*), therefore the available space, *i.e.* surface area of grains, is important. For small grains, the surface-to-volume ratio is higher than for larger grains. This implies a larger surface area

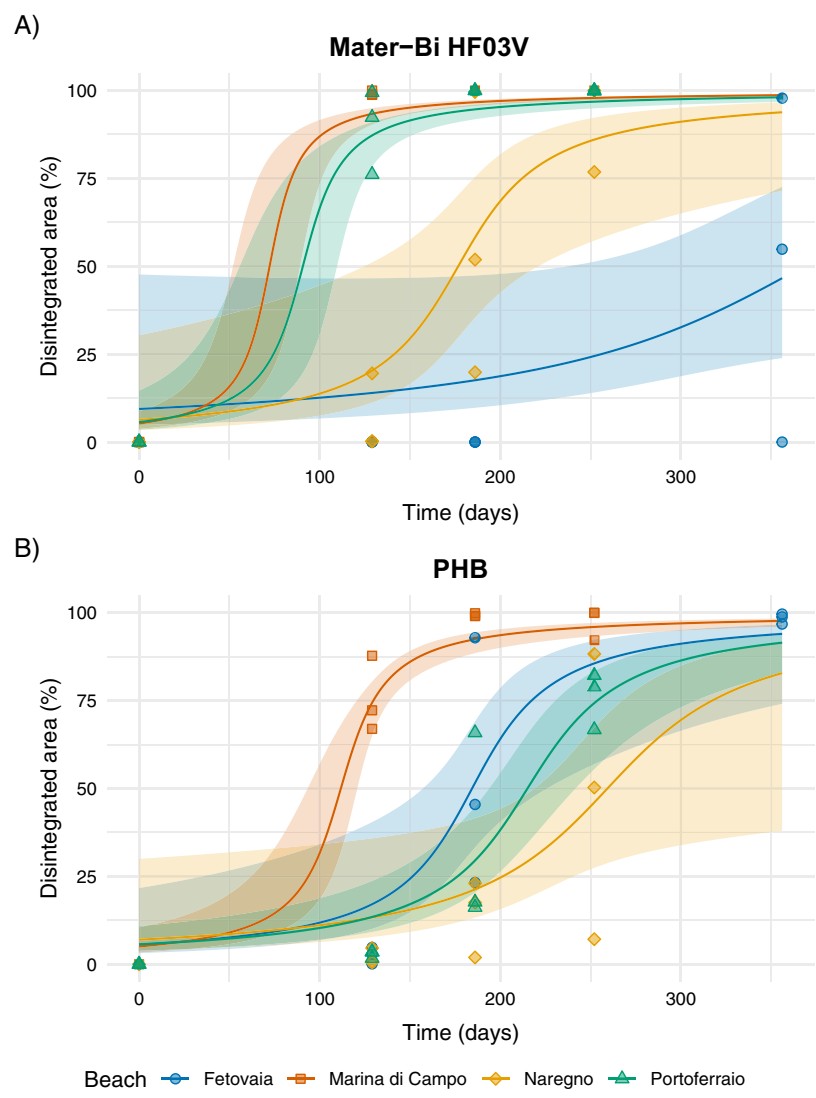

**Figure 5** **(A) Disintegration of Mater-Bi HF03V film (12 μm) and (B) PHB (85 μm) on sediments from four different beaches.** Model curve and 95% confidence interval (shaded area).

available for microbial colonisation and thus a negative relationship between grain size and bacterial abundance (*Rublee & Dornseif, 1978*). Furthermore, during the sieving process, fine organic-rich compounds might have been washed out from the bulk sediment into the finer fractions, which would increase bacterial abundance and activity (*Cammen, 1982*; *Rublee, 1982*; *Stoeck & Albers, 1999*).

These observations for manipulated sediment led to a follow-up experiment in which the influence of natural beach sediment, containing a mix of different grain sizes, on the disintegration of biodegradable plastic was assessed. Here, additionally to Mater-Bi HF03V, a positive (PHB) and negative control (HD-PE) was tested.

Overall, the half-life of PHB (112 to 259 days) was in a similar range as in a previous study with similar settings. *Lott et al. (2021)* tested the same PHB material

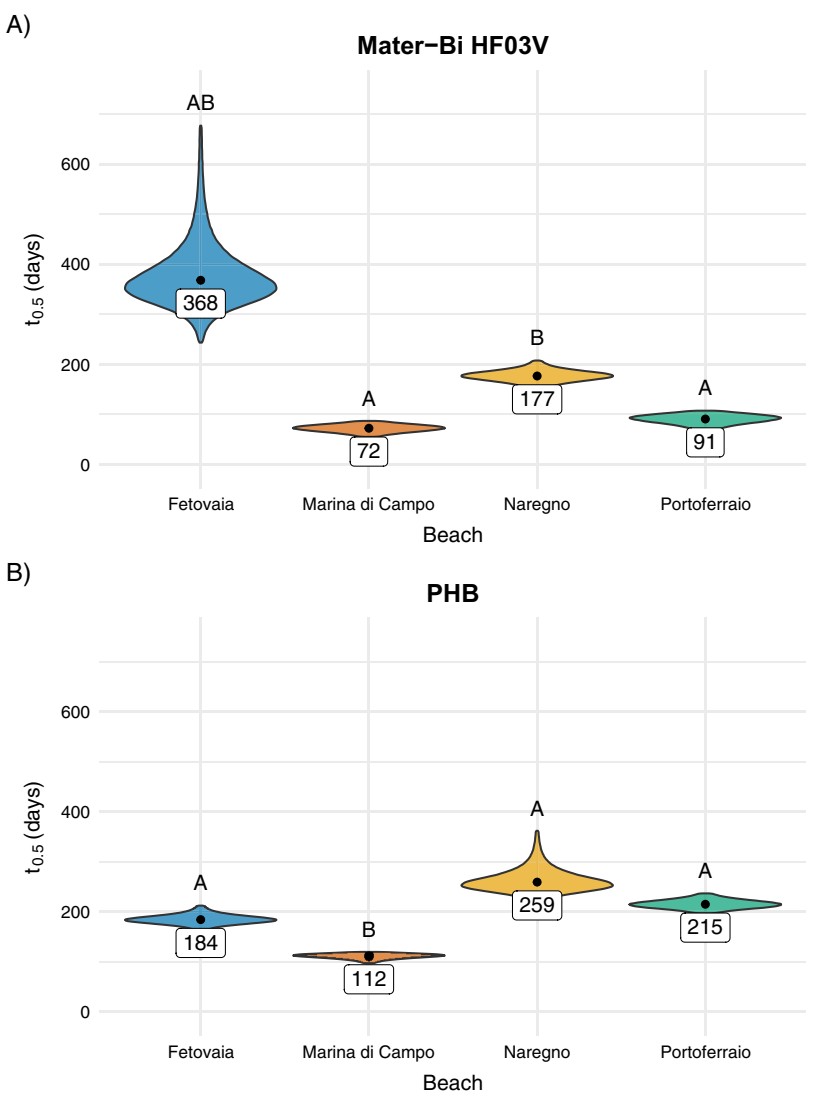

**Figure 6 (A) Predicted half-life of Mater-Bi HF03V film (12 μm) and (B) of PHB (85 μm) on sediments from different beaches.** The distributions of the 95% confidence intervals of 500,000 calculated half-life values are shown as violin plots. Significantly different half-life values (empirical *p*-values) have different letters above the plots. The number below the point is the predicted half-life.

on Mediterranean benthic calcareous sediment in laboratory, mesocosm, and field experiments. The predicted half-live was 116 days in the laboratory experiment (at constantly 20 °C), 357 days in the mesocosm experiment (at constantly 20.5 °C), and 654 days in the field experiment (at 14–20 °C). The predicted half-life in an additional field experiment on calcareous sediment in tropical SE Asia (at a mean temperature of 28.6 °C) was 54 days. This confirms the test conditions applied here as environmentally relevant.

The disintegration of a not further specified type of Mater-Bi was tested in a benthic field experiment off Isola d'Elba at a depth of 36 m (*Pauli et al., 2017*). Disintegration after one year was lower (mean 13%, all replicates below 30%) and less variable than in this study. The temperature in our study was probably more stable than in the field test with

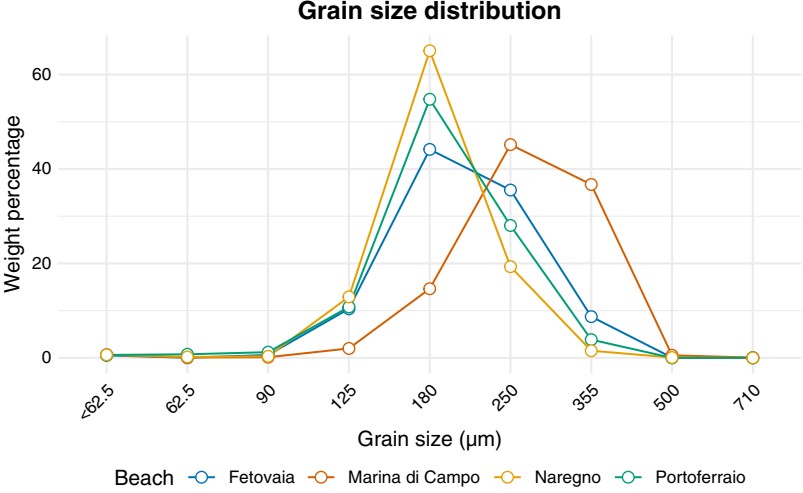

**Figure 7 Grain size distribution of sediment from different beaches.** The weight percentage (y-axis) of each size fraction (x-axis) is shown for sediment from different beaches (colour-coded).

**Table 2 Results of sediment analysis.**

| Parameter | Unit | Fetovaia | Marina di Campo | Portoferraio | Naregno |
|---|---|---|---|---|---|
| Dry matter | % | 74.3 | 72 | 74.6 | 75.2 |
| LOI 550 °C | % DM | 1.0 | 0.6 | 1.1 | 1.1 |
| ROI 800 °C | % DM | 98.6 | 98.9 | 98.1 | 98.4 |
| Nitrogen (Total-N) | mg kg$^{-1}$ DM | <200 | <200 | <200 | <200 |
| Phosphorous | mg kg$^{-1}$ DM | 110 | 110 | 190 | 80 |
| TC | % DM | 0.14 | <0.1 | 0.18 | 0.11 |
| TOC | % DM | <0.1 | <0.1 | <0.1 | <0.1 |
| Organic matter | % DM | 1.0 | 0.6 | 1.1 | 1.1 |
| Calcium carbonate | % DM | 0.9 | 1.1 | 1.8 | 1.1 |
| Silicate | % DM | 98.1 | 98.3 | 97.1 | 97.8 |
| Mean grain size | μm | 245.2 | 322.9 | 232.1 | 220.5 |
| Mean grain size | Description | Fine sand | Medium sand | Fine sand | Fine sand |
| Sorting | | 1.355 | 1.333 | 1.312 | 1.264 |
| Sorting | Description | Well sorted | Well sorted | Well sorted | Very well sorted |
| Skewness | | 0.072 | −0.062 | 0.071 | 0.077 |
| Skewness | Description | Symmetrical | Symmetrical | Symmetrical | Symmetrical |

**Note:**
One sample per sediment type was analysed. LOI, loss in ignition; ROI, residue on ignition; DM, dry matter; TC, total carbon; TOC, total organic carbon.

positive effects on biodegradation rates. No other studies investigated the degradation of Mater-Bi HF03V under marine conditions. However, based on laboratory tests investigating the effect of temperature on the biodegradation rates in soil (*Pischedda, Tosin & Degli-Innocenti, 2019*), formulas are given to calculate the theoretical biodegradation time needed under different temperatures for samples with a given surface area and weight. These formulas can be adapted to calculate the theoretical half-life of samples with the dimensions

used in this study. For the pilot experiment, the theoretical half-life is 28 days at 20 °C and for the follow-up experiment, using thinner films, 13 days. Mater-Bi HF03V exposed to soil seems to degrade much faster than exposed to the marine sediments used in this study with a minimal half-life of 153 days in the pilot experiment and 72 days in the follow-up experiment. Also, other studies observed lower biodegradation rates under marine conditions compared to terrestrial settings (*Chamas et al., 2020*).

PHB disintegrated significantly faster on Marina di Campo sediment than on the other sediment types. Also, Mater-Bi HF03V disintegrated fastest on Marina di Campo sediment, but similarly fast on Portoferraio sediment. Different than expected from the results of the pilot study, the disintegration was faster for the coarser Marina di Campo sediment. The grain size characteristics of the Portoferraio sediment was not substantially different from those of Naregno and Fetovaia sediment (Fig. 6, Tables SI-4 to SI-6). This implies that other environmental factors were crucial and masked the effect of the grain size. For instance, the organic matter content has a strong positive impact on the bacterial abundance (*Cammen, 1982*; *Rublee, 1982*; *Stoeck & Albers, 1999*). However, the organic carbon content in all sediment types was below the detection limit of 0.1% dry mass. Also, when applying the loss-on-ignition method, similar values were calculated for the organic matter content in all sediments (0.6–1.1%). No statistical tests could be performed due to the lack of replication. The phosphorous content in the Portoferraio sediment was slightly higher than in the other sediment types. Phosphorous concentrations from all sites were low compared to a study in the Mediterranean Sea in Turkey (*Gunduz et al., 2011*). The relatively higher concentration of phosphorous in sediment from Portoferraio, the largest town on the island, probably resulted from sewage or industrial effluents (*Berthold et al., 2018*; *Howell, 2010*). Also, Marina di Campo is a small town with more inhabitants than the villages Fetovaia and Naregno, therefore an anthropogenic impact on the bacterial abundance or activity is probable and could explain the faster disintegration. A microbial community presumably exposed to a wider range of organic substances *e.g.* in municipal waste waters from Marina di Campo and Portoferraio may be more prepared or pre-adapted for the degradation of a more complex blend than the community on beaches low in anthropogenic organics input. A more in-depth study, comparing the composition and metabolic functionality of each of the beach sediments could help to explain the differences in the biologically driven disintegration of biodegradable plastic materials observed in this study.

The disintegration of Mater-Bi HF03V on Fetovaia sediment and PHA on Nargeno sediment was highly variable. This could be caused by processes occurring during material disintegration. The increasing biodegradation weakens the structure of the film until reaching an unstable equilibrium. Then, even small differences in biodegradation rates have strong effects on the physical integrity of the sample and therefore its disintegration.

## CONCLUSIONS

From the pilot experiment, we conclude that in general there is a strong impact of sediment grain size on the disintegration rate. In natural beach sediment containing a mix of different grain sizes, no such clear effects could be observed and differences in material

disintegration could be masked by additional environmental parameters. The impact of anthropogenically influenced sediment on the biodegradation rates of biodegradable polymers should be further investigated.

## ACKNOWLEDGEMENTS

We thank D. Makarow and B. Unger for technical help with the experiments.

### Funding

Novamont S.p.A, Novara, Italy funded this study and provided the polymer samples. The funders had no role in study design, data collection and analysis, decision to publish, or preparation of the manuscript.

### Grant Disclosures

The following grant information was disclosed by the authors:
Novamont S.p.A, Novara, Italy.

### Competing Interests

Novamont S.p.A. (Novara, Italy) funded the study. No influence on the design and analysis of the experiments nor on the manuscript was taken by Novamont. Andreas Eich, Miriam Weber, and Christian Lott are employed by HYDRA Marine Sciences GmbH.

### Author Contributions

- Andreas Eich performed the experiments, analyzed the data, prepared figures and/or tables, authored or reviewed drafts of the paper, and approved the final draft.
- Miriam Weber conceived and designed the experiments, performed the experiments, authored or reviewed drafts of the paper, and approved the final draft.
- Christian Lott conceived and designed the experiments, performed the experiments, authored or reviewed drafts of the paper, and approved the final draft.

### Data Availability

  The photos used to assess disintegration and the tables with the analysed disintegrated film area are available in the Supplementary Files.

### Supplemental Information

Supplemental information for this article can be found online at http://dx.doi.org/10.7717/peerj.11981#supplemental-information.

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
