# Peer review of "Disintegration half-life of biodegradable plastic films on different marine beach sediments"

_PeerJ, doi:10.7717/peerj.11981_

## Round 0.1 · original submission · Major Revisions

Please revise your manuscript considering constructive comments from reviewers, and resubmit it.

Reviewer 1 ·

Basic reporting

1. The novelty of this study is suggested to restate. There are many papers addressing biodegradability of the biodegradable plastics under different conditions (e.g. composting, seawater, anaerobic digestion etc.). Did any other researchers investigate the material of Mater-Bi HF03V in the marine conditions? If done, what are their results? And also what is the research gap proposed to address in this study?
2. The background information of tested materials Mater-Bi HF03V is suggested to clearly introduce, including how it was made, what is the molecular weight and characteristics, and why it can replace the traditional plastics, etc.
3. The title of the paper is suggested to rewrite to reflect the specific topic studied in this paper, e.g. the new materials name, and the biodegradation conditions (e.g. grain size).

Experimental design

1.Can the authors list the information of the tested materials in a table and briefly presented in the materials section? it is hard to read the comprehensive information in the text and putting the information in the tables could help the reader a lot understand it.
2. The description of methods including the experimental procedure is suggested to be more detailed, which helps the readers repeat the experiments. Also, the rationale of using such methods are also suggested to present.

Validity of the findings

1. This paper lacks some in-depth discussion of the results, like compared to biodegradation of this material under other environments, the mechanism of biodegradation and involved microorganisms.
2. Why time of 50% degradation was specially investigated in this study? Usually, many researchers and standards look for the time of 90% degradation, like ASTM D6400.

Additional comments

1. The supplemental materials are suggested to have more organization and all the tables and figures are suggested to combined in a file, instead of 14 individual files in this paper. In addition, the titles and oher necessary part were lacked in supplemental materials.

·

Basic reporting

Line 80: May need a reference?

Supplementary tables: Need titles - slightly confusing when going back and forth without any information on the tables.

Raw data table for follow-up study: LDPE should be HDPE?

Fig. SI-5: Temperature log for Naregno missing?

This one might just be a personal preference, but perhaps p-values should be displayed explicitly when significant?

Experimental design

Line 162-179: Is there a particular reason why different sediment layer thickness and strip dimensions were used here compared to the pilot? Could these also be confounding factors impacting the resulting disintegration rates (e.g., due to SA:V ratio)? Considering the distinct difference in results between the pilot and follow-up study, a little justification for these differences may be warranted? Also, is it really just one (not two) replicate? From what I understand from the tables and figures, it seems n=3.

Line 163-165: What is the purpose of sealing with plastic film? Did the authors do the same for the pilot?

Line 180-182: Might want to justify why last sampling for one of the sites was different than the others.

Validity of the findings

Line 219-222: Interesting result but was not further discussed - authors should provide potential justification for this observation? Also, is there an issue of pseudoreplication?

Additional comments

Overall well written piece; very easy to follow and understand. Tables and figures succinctly presented, adequate amount of background information and references provided.

Research question and aim clearly defined. Adequate replication and novel results provided with relevant discussion and sound recommendation and conclusion. Experimental method sufficiently detailed for replication and understanding, but minor additions to explain certain decisions/results may be helpful.

---

## Round 0.2 · accepted · Accept

The authors revised the manuscript considering all suggestions and comments, and it can be accepted in the current version.

Reviewer 1 ·

Basic reporting

The comments have been well addressed.

Experimental design

The comments about the experiment design have been well addressed.

Validity of the findings

The comments about the findings validity have been well addressed.